# Developmental Neurotoxicity of Trichlorfon in Zebrafish Larvae

**DOI:** 10.3390/ijms241311099

**Published:** 2023-07-04

**Authors:** Qipeng Shi, Huaran Yang, Yangli Chen, Na Zheng, Xiaoyu Li, Xianfeng Wang, Weikai Ding, Bangjun Zhang

**Affiliations:** 1Henan International Joint Laboratory of Aquatic Toxicology and Health Protection, College of Life Science, Henan Normal University, Xinxiang 453007, China; 2004183103@stu.htu.edu.cn (H.Y.); chen2787657141@163.com (Y.C.); 041035@htu.edu.cn (X.L.); 2022207@htu.edu.cn (W.D.); zhangbangjun@htu.edu.cn (B.Z.); 2State Key Laboratory of Freshwater Ecology and Biotechnology, Institute of Hydrobiology, Chinese Academic of Sciences, Wuhan 430072, China; zhengna@ihb.ac.cn; 3College of Fisheries, Henan Normal University, Xinxiang 453007, China; wangxianfeng@htu.edu.cn

**Keywords:** trichlorfon, developmental neurotoxicity, neurotransmitter system, central nervous system, zebrafish embryos/larvae

## Abstract

Trichlorfon is an organophosphorus pesticide widely used in aquaculture and has potential neurotoxicity, but the underlying mechanism remains unclear. In the present study, zebrafish embryos were exposed to trichlorfon at concentrations (0, 0.1, 2 and 5 mg/L) used in aquaculture from 2 to 144 h post fertilization. Trichlorfon exposure reduced the survival rate, hatching rate, heartbeat and body length and increased the malformation rate of zebrafish larvae. The locomotor activity of larvae was significantly reduced. The results of molecular docking revealed that trichlorfon could bind to acetylcholinesterase (AChE). Furthermore, trichlorfon significantly inhibited AChE activity, accompanied by decreased acetylcholine, dopamine and serotonin content in larvae. The transcription patterns of genes related to acetylcholine (e.g., *ache*, *chrna7*, *chata*, *hact* and *vacht*), dopamine (e.g., *drd4a* and *drd4b*) and serotonin systems (e.g., *tph1*, *tph2*, *tphr*, *serta*, *sertb*, *htrlaa* and *htrlab*) were consistent with the changes in acetylcholine, dopamine, serotonin content and AChE activity. The genes related to the central nervous system (CNS) (e.g., *a1-tubulin*, *mbp*, *syn2a*, *shha* and *gap-43*) were downregulated. Our results indicate that the developmental neurotoxicity of trichlorfon might be attributed to disorders of cholinergic, dopaminergic and serotonergic signaling and the development of the CNS.

## 1. Introduction

O,O-dimethyl (1-hydroxy 2,2,2-trichloroethyl) phosphonate (Trichlorfon, TCF), a selective organophosphorus pesticide (OP), is widely used in aquaculture to fight against parasites [1,2]. Farmers use TCF to control two common ectoparasites, and the effect is very good but is often used repeatedly and in high concentrations without special guidance. The concentration of trichlorfon used in aquaculture is usually 0.1–1 mg/L, even as high as 25 g/L for the treatment of fish diseases [3]. Repeated and excessive use of trichlorfon poses potential risks to the environment and nontarget species (especially fish) [4]. Trichlorfon can also pollute surface water and groundwater through pesticide application, volatilization and underground infiltration, which poses a threat to the ecological environment, drinking water safety and the survival of nontarget organisms in the water source area [5]. According to the Environmental Protection Agency (2009) report, the concentrations of trichlorfon are 0.00027 and 0.179 mg/L in underground water and surface water, respectively [6]. The trichlorfon concentration was detected in the surface water of the Dongjiang River at up to 50 ng/L, and the risk coefficient was higher than 0.1, suggesting a potential risk to aquatic organisms [7,8].

Several studies have shown that trichlorfon has immunotoxicity [9,10,11], oxidative stress [5,6,11], hepatotoxicity [12] and hematotoxicity [13,14,15] in fish. Its potential neurotoxicity has attracted much attention, as a recent study reported that trichlorfon can cross the blood–brain barrier [16] and accumulate in the brains of fish, indicating that the brain is one of the potential target organs of trichlorfon. In addition, a recent study reported that trichlorfon has adverse effects on the behavior of silver catfish [17], which has been widely regarded as an important indicator of neurodevelopment. Therefore, this study suggests that trichlorfon has potential neurotoxicity.

The main toxic mechanism of OP inhibits AChE activity [18,19], and exposure to trichlorfon causes significant inhibition of AChE in fish [11,16,17,20]. This suggests that trichlorfon-induced neurotoxicity may be associated with the cholinergic system. However, whether trichlorfon can affect other neurotransmitter systems (e.g., dopaminergic and serotonergic) is unclear. Cholinergic, dopaminergic and serotonergic systems are chemical messengers that are responsible for information transmission between neurons in the central nervous system (CNS) [21,22,23,24] and have been widely used as biomarkers for neurotoxicity evaluation [25,26,27]. Thus, we hypothesized that the neurotoxicity induced by trichlorfon is related to neurotransmitter systems.

Therefore, we used zebrafish embryos as the teratotoxicity model to investigate the role of neurotransmitter signals (cholinergic, dopaminergic and serotonergic) in the neurotoxicity induced by trichlorfon. The transcription levels of several key genes related to CNS are often regarded as biomarkers for evaluating neurotoxicity and behavioral changes [26]. AChE has been widely used as a biomarker for neurotoxins such as organophosphorus pesticides [19]. In this study, we investigated (1) locomotor behaviors, (2) the binding energy and binding mode simulated by molecular docking, (3) certain neurotransmitter (acetylcholine, dopamine and serotonin) contents and AChE activity and (4) several key genes associated with the development of the CNS and cholinergic, dopaminergic and serotonergic systems.

## 2. Results

### 2.1. Molecular Docking with AChE

Based on the results of molecular docking, trichlorfon could dock into the active sites of AChE (Figure 1A), which is similar to huprine W (an AChE inhibitor) binding with human AChE. Trichlorfon formed two hydrogen bonds with SER147 and TYR355 residues in the active pocket of AChE (Figure 1B). The lowest binding energy between trichlorfon and AChE was −18.8 kJ/mol.

### 2.2. Developmental Parameters

Trichlorfon exposure caused obvious developmental toxicities in zebrafish larvae at 144 hpf. Reduced survival rate, hatching rate, heart rate and body length and increased malformation rate are shown in The survival rate and hatching rate were reduced by 6.22% and 7.26%, respectively, in 5 mg/L of trichlorfon (Table 1). The malformation rates (Appendix A; spinal curvature, tail curvature and yolk sac edema) were significantly increased in the 2 mg/L (121.91%; *p* < 0.05) and 5 mg/L (652.30%; *p* < 0.001) exposure groups (Table 1). The heart rate was decreased by 8.17% (*p* < 0.001) in the 5 mg/L of trichlorfon exposure group (Table 1). Exposure to 5 mg/L of trichlorfon decreased the body length of larvae by 10.18% (*p* < 0.001) (Table 1). There was no significant change in body weight compared to the control group.

### 2.3. Locomotor Behavior of Zebrafish Larvae

The locomotor activity of the zebrafish larvae at 144 hpf was assessed with a continuous light for 10 min and a 30 min dark-to-light stimulation program (Figure 2). Exposure to 5 mg/L of trichlorfon significantly reduced the mean locomotor speed of the larvae under the continuous light stimulation period (*p* < 0.01; Figure 2A). During the dark–light transition period, locomotor traces of the larvae were observed (Figure 2C). In the first dark (*p* < 0.05; Figure 2B) and second dark phase (*p* < 0.01; Figure 2B), the mean locomotor speed of the larvae was significantly reduced in the 5 mg/L exposure group. In the third dark phase, the mean locomotor speed of the larvae was increased in the 5 mg/L exposure group (*p* < 0.05; Figure 2B). In the first light phase, a significant decrease in larval swimming speed was observed upon exposure to 2 mg/L (*p* < 0.01; Figure 2B) and 5 mg/L (*p* < 0.01; Figure 2B) of trichlorfon. In the second light phase, the mean locomotor speed of the larvae was significantly reduced in the 5 mg/L group (*p* < 0.001; Figure 2B).

### 2.4. Gene Expression

The transcription of several key genes associated with CNS development was observed after exposure to trichlorfon (Figure 3). These genes included *α1-tubulin*, myelin basic protein (*mbp*), synapsin IIa (*syn2a*), sonic hedgehog a (*shha*) and growth-associated protein 43 (*gap-43*). Significant downregulation of α1-tubulin and mbp was observed by 1.71- and 1.78-fold, respectively, at 0.1 mg/L of trichlorfon, by 1.96- and 3.49-fold, respectively, at 2 mg/L of trichlorfon and by 2.54- and 1.70-fold, respectively, at 5 mg/L of trichlorfon (Figure 3). *Shha* transcription was significantly downregulated by 2.09-fold, 2.29-fold and 3.28-fold at 0.1 mg/L and 2 mg/L of trichlorfon, respectively (Figure 3). *Syn2a* and *gap-43* were significantly downregulated by 2.20- and 1.92-fold, respectively, at 2 mg/L of trichlorfon and by 2.10- and 1.98-fold, respectively, at 5 mg/L of trichlorfon (Figure 3).

The expression levels of genes related to the cholinergic, dopaminergic and serotonergic systems after exposure to trichlorfon are shown in Figure 3. Cholinergic system genes included acetylcholinesterase (*ache*), nicotinic acetylcholine receptor α7 subunit (*chrna7*), choline acetyltransferase a (*chata*), high-affinity choline transporter (*hact*) and vesicular acetylcholine transporter (*vacht*). The downregulation of ache was observed by 1.67-, 1.86- and 2.64-fold at 0.1, 2, and 5 mg/L of trichlorfon, respectively (Figure 3). Chrna7, hact and vacht were significantly downregulated by 3.48-, 8.73- and 3.68-fold at 2 mg/L and by 7.18-, 31.34- and 5.47-fold at 5 mg/L of trichlorfon (Figure 3). Chata was significantly upregulated by 1.38-fold after exposure to 5 mg/L of trichlorfon (Figure 3). Dopaminergic system genes included mesencephalic astrocyte-derived neurotrophic factor (*manf*), brain-derived neurotrophic factor (*bdnf*), nuclear receptor subfamily 4, group A, member 2b (*nr4a2b*) and dopamine receptor (*drd2b, drd4a, drd4b* and *drd7*). The downregulation of drd4a and drd4b was observed by 2.77- and 2.26-fold at 0.1 mg/L and by 4.99- and 3.67-fold at 5 mg/L of trichlorfon, respectively (Figure 3). There were no significant differences in *manf*, *bdnf*, *nr4a2b*, *drd2b* and *drd7* between the exposure groups and the control group (Figure 3). Serotonergic system genes included tryptophan hydroxylase (*tph1*, *tph2* and *tphr*), serotonin transporter (*serta* and *sertb*) and 5-hydroxytryptamine receptor (*htr1aa* and *htr1ab*). The upregulation of tph1 was observed by 2.11-fold after exposure to 5 mg/L of trichlorfon (Figure 3). *Tph2* and *thpr* were significantly upregulated by 1.76- and 2.47-fold, respectively, at 2 mg/L of trichlorfon and by 1.98- and 2.88-fold, respectively, at 5 mg/L of trichlorfon (Figure 3). The downregulation of serta was observed by 3.70- and 6.57-fold after exposure to 2 and 5 mg/L of trichlorfon, respectively (Figure 3). *Sertb*, *htr1aa* and *htr1ab* were significantly downregulated by 2.32-, 2.51- and 1.84-fold, respectively, at 0.1 mg/L of trichlorfon, by 2.51-, 5.08- and 1.73-fold at 2 mg/L of trichlorfon and by 11.28-, 24.01- and 7.15-fold at 5 mg/L of trichlorfon (Figure 3).

### 2.5. Neurotransmitter Content

We examined the concentrations of ACh, DA and 5-HT (Figure 4). The ACh content was significantly increased by 18.39% and 55.67% at 2 mg/L and 5 mg/L of trichlorfon, respectively (Figure 4A). The DA content was significantly increased by 45.48% in the 5 mg/L group (Figure 4B). The content of 5-HT was significantly increased by 13.67% and 50.39% at 2 mg/L and 5 mg/L of trichlorfon, respectively (Figure 4C).

### 2.6. AChE Activity

At 144 hpf, the total AChE activities of the zebrafish larvae were measured. Significant inhibition of AChE activities was induced by 25.59%, 55.26% and 85.17% at 0.1 mg/L, 2 mg/L and 5 mg/L of trichlorfon, respectively (Figure 5).

## 3. Discussion

In our study, the developmental neurotoxicity of trichlorfon was assessed by using zebrafish larvae as a model, as the brains of zebrafish in the early developmental stages are vulnerable to damage by toxicants [28,29]. A decrease in larval behavior was observed after exposure to trichlorfon. The significant inhibition of AChE activity, the altered contents of three neurotransmitters (ACh, DA and 5-HT) and the altered expression of key genes related to CNS development and neurotransmitter (ACh, DA and 5-HT) signaling indicated that trichlorfon-induced neurotoxicity might be assigned to disorders of the cholinergic, dopaminergic and serotonergic systems and neuronal development in zebrafish larvae.

The developmental toxicity of zebrafish larvae was observed upon exposure to trichlorfon, including a reduced survival rate, hatching rate, heart rate and body length, as well as an increased malformation rate. In this study, exposure to trichlorfon from 2 to 144 hpf significantly decreased the survival rate, hatching rate, heart rate and body length of zebrafish larvae at 5 mg/L and increased the malformation rate at 2 and 5 mg/L. A study showed that exposure to trichlorfon significantly reduces the survival rate (up to 40 mg/L) and increases the malformation rate (20 mg/L) of zebrafish embryos/larvae at 4 days and reduces the hatching rate (40 mg/L) of zebrafish embryos/larvae at 3 days [30]. Previous studies on the effects of trichlorfon on heart rate and body length have not been reported. Likewise, a recent study reported that exposure to other OPs, such as chlorpyrifos, decreases the survival rate (up to 1500 nM), hatching rate (up to 1500 nM) and heart rate (up to 1000 nM) and increases the malformation rate (up to 1000 nM) [31]. Moreover, Altenhofen et al. (2019) reported that exposure to 5 and 10 mg/L dichlorvos, the metabolite of trichlorfon, from 1 hpf to 7 dpf significantly decreases the heart rate and body length of zebrafish [32]. Taken together, the significant impairment of developmental indexes observed in this study indicate moderate developmental toxicity of trichlorfon compared to other OPs.

The locomotor behavior of zebrafish larvae has been widely regarded as an important index of neurodevelopment [33,34]. Hypoactivity of larvae was observed upon exposure to trichlorfon in this study, indicating the neurotoxicity of trichlorfon. Recently, a study revealed abnormal behavior of juvenile silver catfish exposed to trichlorfon (11 mg/L) for 48 h [17]. Likewise, exposure to other OPs (diazinon, dichlorvos, malathion and methyl-parathion) induces abnormal zebrafish behavior [35]. Similarly, a study conducted by Serafini et al. (2019) revealed that exposure to the pesticide eprinomectin affects the locomotor behavior of silver catfish [36]. Changes in locomotor behavior may be the result of changes in AChE activity, the transmission of neurotransmitter signals and the development of the neuron [34,37]. Therefore, we further investigated the effects of trichlorfon on AChE activity, neurotransmitter systems and gene expression related to CNS.

Molecular docking was applied to probe the interaction between trichlorfon and AChE. The results of molecular docking showed that trichlorfon could bind with AChE, suggesting that trichlorfon has the potential to affect AChE activity. Moreover, in this study, the AChE activity of zebrafish larvae was significantly inhibited after exposure to trichlorfon at concentrations of 0.1, 2 and 5 mg/L. AChE has been widely regarded as an important biomarker for OP exposure [19,38,39]. Several studies have indicated that exposure to trichlorfon can inhibit AChE activity in various fish [11,17,40]. For example, a recent study showed that exposure to trichlorfon (0.1, 0.5 and 1 mg/L) for 28 days can significantly inhibit AChE activity in the brain of the common carp [11]. Thus, consistent with these previous studies, the present study suggests that trichlorfon may be a potent AChE inhibitor. AChE is also involved in the development of the CNS and neural regeneration in the early life stages of zebrafish embryos [41]; thus, the inhibition of AChE activity might interfere with neuronal development.

In addition, AChE can hydrolyze ACh to Ch when released into the cholinergic synapse [42,43]. Therefore, the inhibition of AChE activity contributed to the increased ACh content in this study. Moreover, because ACh is involved in motor (voluntary movements) and learning memory behaviors [24,44], the increase in ACh content may cause behavioral changes in this study. Finally, the transcription of genes related to cholinergic signaling, e.g., *ache*, *chrna7*, *chata*, *hact* and *vacht,* was detected. The expression of *ache*, *chrna7*, *hact* and *vacht* was significantly downregulated, and that of *chata* was significantly upregulated after trichlorfon treatment. In the terminals of cholinergic neurons, ACh is synthesized by choline acetyltransferase (ChAT). Subsequently, ACh is transferred to vesicles to form the vesicular acetylcholine transporter (VAChT) and is released into the synaptic space [45]. In the prominent space, ACh interacts with the ACh receptor of the postsynaptic membrane (e.g., 7nAChR). ACh released into the synaptic space is hydrolyzed into choline and acetic acid by acetylcholinesterase (AChE) [45]. Ch is carried into presynaptic neurons via the high-affinity choline transporter (HACT). Hydrolyzed Ch can return to the synaptic terminals and participate in the synthesis of ACh again [45]. Additionally, when ACh content increased, 7nAChR, HACT and VACHT were reduced, and ChAT increased upon exposure to environmental toxins [44]. In our study, decreased ACh content, the downregulation of chrna7, hact and vacht and the upregulation of chata were consistent with this phenomenon. The downregulation of the ache gene was also accompanied by the inhibition of AChE activity in this study. Moreover, these genes are related to locomotor behavior, learning and memory [46]; thus, altering these genes could have resulted in behavioral changes in this study.

Dopamine plays a key role in the development of dopaminergic neurons and the regulation of multiple physiological functions of the CNS [47] and is involved in learning, motor and motivational functions [48]. The content of dopamine was significantly increased upon exposure to trichlorfon in our study. It has been shown that other environmental contaminants, such as TDCIPP, DE-71 and the organophosphorus herbicide glyphosate, can affect dopamine content in zebrafish [49,50,51]. Taken together, these studies indicate that dopamine is vulnerable to exposure to contaminants. Moreover, the expression of *drd4a* and *drd4b*, which are related to dopaminergic signaling, was significantly downregulated in our study. *Drd4a* and *drd4b* are involved in the regulation of dopamine content and play biological roles by activating inhibitory G protein (Gi) [52]. Thus, the downregulation of *drd4a* and *drd4b* observed in the present study might lead to increased dopamine content. *Drd4a* and *drd4b* were reported to be associated with the cognitive function of zebrafish and can mediate the inhibitory effect of drugs on the locomotor activity of zebrafish larvae [52]. Overall, the downregulation of *drd4a* and *drd4b* might have contributed to the decreased dopamine content in our study, which may have ultimately resulted in the alteration of behavioral activity in zebrafish larvae.

In addition, the 5-HT content in this study was significantly increased after exposure to trichlorfon. 5-HT is highly abundant in cortical matter and synapses, and alterations in 5-HT content can cause anxiety behavior [53,54]. A recent study showed that increased 5-HT content can cause behavioral changes in zebrafish upon exposure to the organophosphorus herbicide glyphosate [49]. Likewise, a previous study showed that altered 5-HT content can cause behavioral changes in zebrafish larvae upon exposure to PBDEs [55]. Thus, the results indicate that the hypoactivity of zebrafish larvae upon trichlorfon exposure is associated with increased 5-HT content. Furthermore, the expression of genes (*tph1*, *tph2*, *tphr*, *serta/b* and *htr1aa/b*) related to the synthesis and neurotransmission of 5-HT was tested. 5-HT is synthesized by tryptophan hydroxylases (e.g., TpH 1, TpH 2 and TpHr) [56]. The expression of *tph1*, *tph2* and *tphr* was significantly upregulated after exposure to trichlorfon in our study. Thus, the upregulation of *tph1*, *tph2* and *tphr* may have led to increased 5-HT content in zebrafish larvae exposed to trichlorfon in our study. Moreover, the downregulation of *serta/b* and *htr1aa/b* genes was also observed in this study. The 5-HT transporter (SERT) is expressed in the CNS of the presynaptic membrane of nerve endings, reuptakes 5-HT from the synaptic gap and enters presynaptic neurons, thereby regulating 5-HT neurotransmission and content [57,58]. Additionally, the knockout of the *sert* gene in rats can alter 5-HT levels in the brain [59,60]. The 5-HT1A receptors (*htr1aa* and *htr1ab*) are inhibitory 5-HT receptors that are present in high densities in the cerebral cortex, hippocampus, septum, amygdala and septal nucleus and in low amounts in the basal ganglia and thalamus [61] and are negative feedback regulators of 5-HT content [57,62]. Thus, the downregulation of *serta/b* and *htr1aa/b* may have resulted in increased 5-HT levels in this study after exposure to trichlorfon.

The altered expression of genes related to the CNS is considered to be an important factor in developmental neurotoxicity [34,37]. Therefore, the transcription of genes associated with the CNS development of neurons was assessed. The expression of the genes *α1-tubulin*, *mbp*, *syn2a*, *shha* and *gap-43* in this study was downregulated. *α1-tubulin* is abundantly expressed in the developing neurons of zebrafish larvae and encodes an intermediate filament protein involved in the assembly of microtubules [63]. The downregulation of *α1-tubulin* may cause abnormalities in microtubule structure and function. *Mbp* is a strong basic membrane protein synthesized by vertebrate CNS oligodendrocytes and peripheral nervous system Schwann cells, which maintains the structural and functional stability of the CNS myelin sheath [63]. The function of *shha* is the regulation of vertebrate organ development as an important morphogenetic factor [64]. Therefore, the downregulation of *shha* upon exposure to trichlorfon may affect the development of the brain and other organs. *Syn2a* plays an important role in neurotransmitter release, synaptogenesis and neuronal differentiation of the CNS [65,66]. Thus, the downregulation of *syn2a* observed in the present study may affect synaptogenesis, neurotransmitter release and neuronal differentiation. *Gap-43* is a neurospecific protein involved in synapse formation, neurite outgrowth and nerve cell regeneration and is expressed at high levels during neuronal development and regeneration [67,68]. In addition, *gap-43* is regarded as a marker of nerve damage caused by toxin exposure [69,70]. Thus, the downregulation of *gap-43* indicated neurological damage in zebrafish larvae after exposure to trichlorfon. Likewise, previous studies have reported that the downregulation of the marker genes *α1-tubulin*, *mbp*, *syn2a*, *shha* and *gap-43* have been observed in zebrafish larvae upon exposure to other toxins, e.g., OPs (chlorpyrifos and carbaryl) [70], OPFRs (TPHP and TDCIPP) [34] and PBDEs [37]. Overall, these studies suggest that alterations in the expression of genes related to the CNS may lead to neurotoxicity through involvement in CNS differentiation, neuron development, cytoskeleton maintenance and synapse formation.

## 4. Materials and Methods

### 4.1. Chemicals and Reagents

Trichlorfon (CAS 52-68-6; 97% purity) was purchased from Jiangshan Agrochemical & Chemicals Limited Liability Co., Ltd. (Nangtong, China). Dimethyl sulfoxide (DMSO) and methanesulfonate (MS-222) were supplied by Sigma-Aldrich (Shanghai, China). Trizol reagent was obtained from Takara (Dalian, China). PrimeScript^®^ reverse-transcription reagent kits and SYBR^®^ Real-time PCR Master Mix-Plus kits were purchased from Yeasen (Shanghai, China). In the present study, all other chemicals used were analytical grade.

### 4.2. Zebrafish Culture and Embryo Exposure

Four-month-old wild-type zebrafish (AB strain) cultures and embryo exposure were carried out according to a published study [34]. In brief, 300 embryos at 2 h post fertilization (hpf) that reached the blastula stage and that had developed normally were placed into glass culture dishes with 200 mL of exposure solutions of 0.1, 2 and 5 mg/L of trichlorfon. The exposure concentrations in the present study were based on concentrations used in aquaculture [3]. The DMSO concentration of the control group and the exposure group was 0.01% (*v*/*v*), and there were four replicates for each group. The embryos were exposed at 144 hpf, as most of the organs of the larvae were fully developed, and larvae were able to swim freely by this time. The exposure solutions were completely renewed every day, and the dead embryos/larvae were removed on time. The embryos/larvae were observed every day, and the survival, hatching and malformation were recorded at 144 hpf. Several zebrafish larvae at 144 hpf were fixed in 1% low-melting-point agarose to count heartbeats for 30 s under a stereomicroscope (OLYMTUS, Tokyo, Japan). We also measured the body weights and body lengths of larvae at 144 hpf. After exposure, six larvae with normal morphological appearances and that were swimming freely were selected from each replicate to measure swimming behavior, and the other larvae were anesthetized with 0.01% MS-222 (100 mg/L) and then stored at −80 °C for gene transcription assays, neurotransmitter content determination and enzyme activity measurement.

### 4.3. Molecular Docking

Molecular docking was performed using AutoDock 4.0 (San Francisco, CA, USA). The 3D molecular structure of trichlorfon was obtained from PubChem as a ligand. Because the zebrafish AChE crystal structure was not found in the protein database, the human AChE crystal 4BDT (AChE complex with huprine W) was downloaded from RCSB PDB (http://www.rcsb.org (accessed on 2 April 2021)) and chosen as the molecular docking receptor, as huprine W is a potent AChE inhibitor that binds with the AChE active site [31], and its structure is similar to that of trichlorfon. Molecular docking between the ligand and receptor was performed with AutoDock 4.0 Vina (San Francisco, CA, USA), and the lowest binding affinity energy was selected as the most likely binding status. Finally, the binding results were imported into the PyMOL v2.4 software (Schrödinger, New York, NY, USA) to view the docking results and draw the image.

### 4.4. Locomotor Behavior

The larval behavior assessment was carried out using the Zebralab Video-Track system (ViewPoint Life Sciences, Civrieux, France), as described in a previous study [37]. Zebrafish larvae at 144 hpf (6 larvae per replicate, n = 4) were placed in 24-well plates, with one larva in each containing 1.5 mL of fresh fish water to measure swimming speed. Before the experiment, larvae were transferred to the system to acclimate for 5 min in advance. The procedure was as follows: continuous light time for 10 min and a light and dark transition cycle of 30 min (5 min of darkness and 5 min of light, three cycles). The distance traveled and total duration of movement data were collected every 30 s.

### 4.5. Quantitative Real-Time Polymerase Chain Reaction (qRT-PCR)

Total RNA extraction, first-strand cDNA synthesis and qRT-PCR were performed following a published protocol [55]. Approximately 20 larvae (n = 4 replicates) were homogenized on ice to extract the total RNA with 1 mL of TRIzol reagent. The total RNA concentration and absorbance at 260/280 nm ratios were determined with a NanoDrop 2000 spectrophotometer (Thermo Fisher Scientific, Waltham, MA, USA). The reverse transcription of RNA was carried out with a PrimeScript^®^ RT Reagent kit (Yeasen, Shanghai, China) following the manufacturer’s instructions. qRT-PCR was analyzed on an ABI 7300 system (Thermo Fisher Scientific) with the SYBR^®^ Real-time PCR Master Mix-Plus kits (Yeasen, Shanghai, China). The PCR program was as follows: 95 °C for 5 min, followed by 40 cycles at 95 °C for 10 s, 66 °C for 20 s and 72 °C for 20 s. The primer sequences of the genes in this study were designed using the online Primer 3 software (http://primer3.ut.ee/ (accessed on 6 September 2022)) and are listed in Appendix A. In our study, *β-actin* was used as the reference gene, as its transcription was stable upon trichlorfon exposure. The relative change in the target gene transcription level was calculated using the 2^−ΔΔCt^ method.

### 4.6. Neurotransmitter Measurements

Approximately 50 larval samples (n = 4 replicates) were homogenized in 300 μL of extracting solution on ice. After centrifugation at 3500 rpm at 4 °C for 15 min, the supernatants were transferred into a new tube for analysis. The contents of ACh, DA and 5-HT were measured using ACh, DA and 5-HT assay kits (Nanjing Jiancheng Bioengineering Institute, Nanjing, China) following the manufacturer’s instructions. The optical densities of ACh, DA and 5-HT were recorded at 550 nm, 450 nm and 450 nm, respectively. The contents of ACh, 5-HT and DA were obtained from the standard curve and are expressed as μg/mgprot, ng/mgprot and pg/mgprot, respectively.

### 4.7. Acetylcholinesterase Activity Measurements

The AChE activity was measured based on the content to which ACh was catalytically hydrolyzed to choline (Ch), as Ch could react with aym-trinitrobenzene (TNB) to form yellow compounds. An amount of 500 μL of normal saline was added to the sample (approximately 40 larvae, n = 4 replicates) and homogenized on ice, and it was then centrifuged at 2500 rpm at 4 °C for 10 min. The supernatants were diluted with normal saline tenfold. The enzyme activity was determined with the AChE assay kit (Nanjing Jiancheng Bioengineering Institute, Nanjing, China) according to the manufacturer’s instructions. The ACh content was measured based on ACh reacting with the substrate solution to form a brown compound through a color reaction. DA and 5-HT contents were measured based on double antibody Sandwich-ELISA. The absorbance was determined at 412 nm, and the enzyme activities were expressed in U/mg.protein. The concentration of the protein was determined based on the BCA method.

### 4.8. Statistical Analysis

SPSS 18.0 software (Chicago, IL, USA) was used for the statistical analysis of experimental data. All data are expressed as the mean ± standard error of the mean (SEM). The Kolmogorov–Smirnov test and Levene test were used to analyze the normal distribution and variance homogeneity of the data. A one-way analysis of variance (ANOVA) was used to compare the differences between the control group and the exposure groups, and Tukey’s multiple comparisons method was used for the P test. *p* < 0.05 indicates a significant difference.

## 5. Conclusions

Overall, these observations suggest that exposure to trichlorfon at concentrations used in aquaculture can lead to neurotoxicity in zebrafish larvae, and this is due to changes in cholinergic, dopaminergic and serotonergic signaling, as well as neuronal development. However, the mechanisms of altered gene expression associated with neurotransmitter systems and CNS development remain unclear. Therefore, more future studies are required to confirm the potential molecular targets of trichlorfon. In addition, because the trichlorfon metabolite dichlorvos is more toxic than itself, more attention should be given to the toxicity of its metabolites.

## Figures and Tables

**Figure 1 ijms-24-11099-f001:**
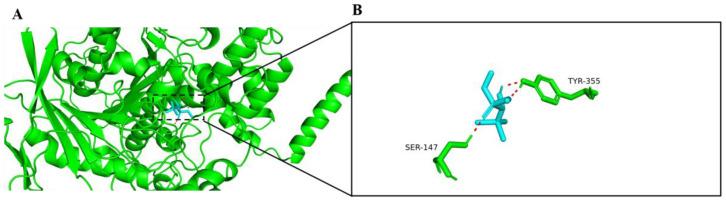
Molecular docking of trichlorfon with AChE. (**A**) Docking profiles between AChE and trichlorfon. Trichlorfon is shown in the figure in the ‘stick’ format, whereas AChE is shown as a 3D cartoon. (**B**) Details of predicted binding mode of AChE and trichlorfon. The contact residues are shown and labeled by type and number. The red dotted line illustrates the hydrogen bond interaction.

**Figure 2 ijms-24-11099-f002:**
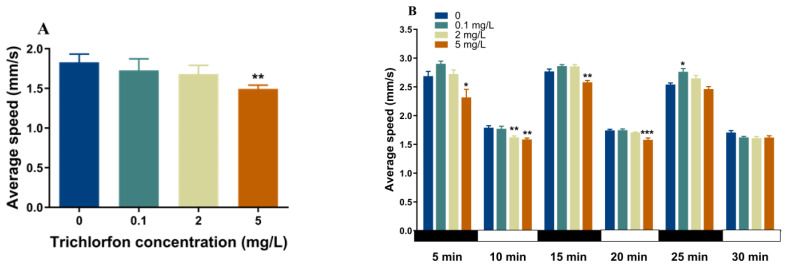
Locomotor behavior of zebrafish larvae after exposure to trichlorfon. Locomotor behavior was assessed in zebrafish larvae exposed to 0, 0.1, 2 and 5 mg/L of trichlorfon until 144 hpf. The following parameters were measured: average swimming speed during a continuous light test (**A**), locomotor traces (**B**) and average swimming speed of the larvae during a light–dark–light–dark photoperiod stimulation test (**C**). Data are expressed as the mean ± SEM value (n = 4 replicates, 6 larvae per replicate) in 30 s intervals. * *p* < 0.05, ** *p* < 0.01 and *** *p* < 0.001 compared to the control group at each time point.

**Figure 3 ijms-24-11099-f003:**
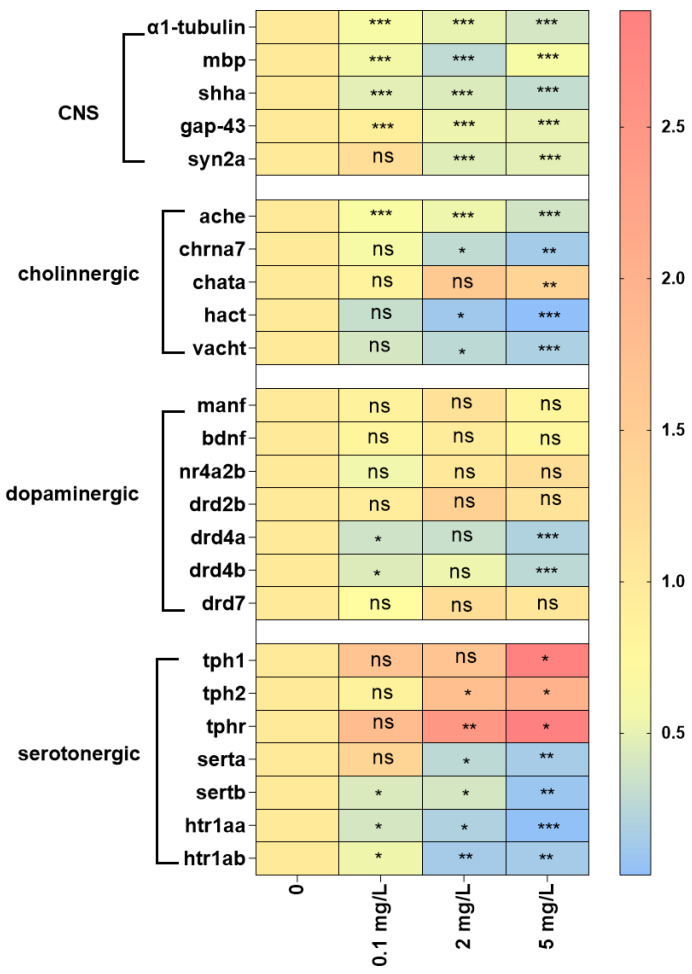
Transcription of CNS development and specific neurotransmitters system-related genes in zebrafish larvae after exposure to trichlorfon. The expression of genes related to CNS development (*a1-tubulin*, *mbp*, *syn2a*, *shha* and *gap-43*), cholinergic system (*ache*, *chrna7*, *chata*, *hact* and *vacht*), dopaminergic system (*manf*, *bdnf*, *nr4a2b*, *drd2b*, *drd4a*, *drd4b* and *drd7*) and serotonergic system (*tph1*, *tph2*, *tphr*, *serta*, *sertb*, *htr1aa* and *htr1ab*) in zebrafish larvae after exposure to various concentrations of trichlorfon (0, 0.1, 2 amd 5 mg/L) until 144 hpf. All data are expressed as the mean ± SEM value (n = 4 replicates, 30 larvae per replicate). * *p* < 0.05, ** *p* < 0.01 and *** *p* < 0.001 compared to the control group. ns: no significant differences.

**Figure 4 ijms-24-11099-f004:**
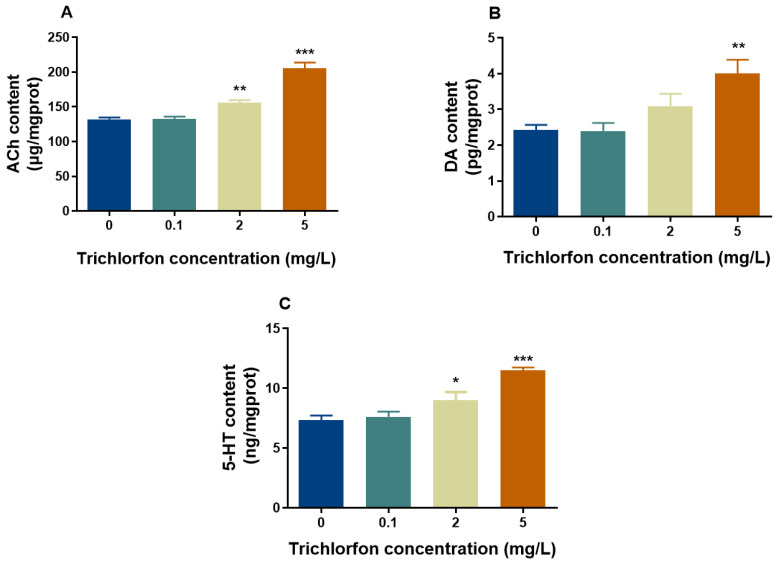
Levels of neurotransmitters in zebrafish larvae exposed to different concentrations of trichlorfon. The contents of certain neurotransmitters, including (**A**) acetylcholine, (**B**) dopamine and (**C**) serotonin upon exposure to 0, 0.1, 2 and 5 mg/L of trichlorfon until 144 hpf, are shown. All data are expressed as the mean ± SEM value (n = 4 replicates, 50 larvae per replicate). * *p* < 0.05, ** *p* < 0.01 and *** *p* < 0.001 compared to the control group. ACh: acetylcholine; DA: dopamine; 5-HT: serotonin.

**Figure 5 ijms-24-11099-f005:**
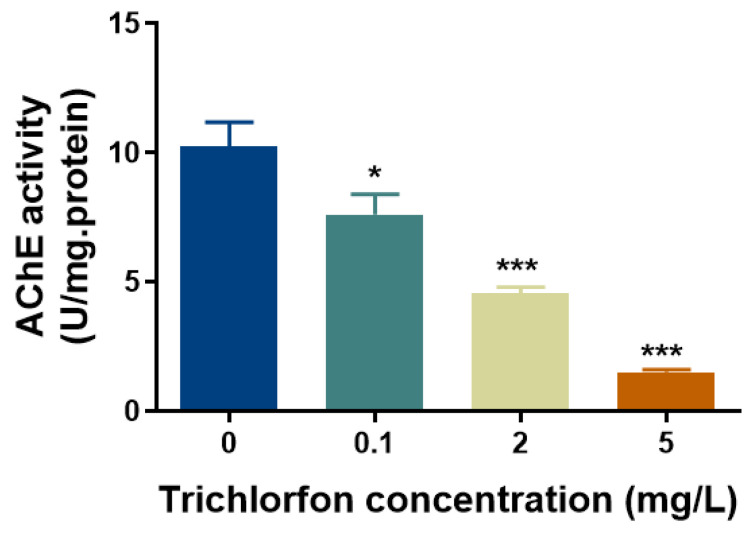
Acetylcholinesterase (AChE) activity in response to trichlorfon. AChE activity in zebrafish larvae was measured following exposure to various concentrations of trichlorfon (0, 0.1, 2 and 5 mg/L) until 144 hpf. All data are expressed as the mean ± SEM value (n = replicates, 40 larvae per replicate). * *p* < 0.05 and *** *p* < 0.001 compared to the control group.

**Table 1 ijms-24-11099-t001:** Developmental toxicity of exposure to trichlorfon for 144 h in zebrafish larvae.

Trichlorfon (mg/L)	Survival Rate (%)	Hatching Rate (%)	Malformation Rate (%)	Heart Rate (Beats/min)	Body Length (mm)	Body Weight (mg/Larvae)
0	90.31 ± 0.79	90.31 ± 0.79	2.83 ± 0.56	170.20 ± 2.59	3.93 ± 0.11	0.56 ± 0.01
0.1	89.56 ± 0.67	88.75 ± 0.88	3.28 ± 0.42	169.00 ± 1.98	3.84 ± 1.63	0.56 ± 0.01
2	88.13 ± 0.36	88.13 ± 0.36	6.28 ± 0.73 *	164.80 ± 1.45	3.72 ± 0.12	0.55 ± 0.01
5	84.69 ± 1.39 **	83.75 ± 0.88 ***	21.29 ± 1.37 ***	156.3 ± 1.37 ***	3.53 ± 0.13 ***	0.50 ± 0.01

Values represent mean ± SEM (n = 4 replicates, 100 larvae per replicate). Significant differences from the control are indicated by * *p* < 0.05, ** *p* < 0.01 and *** *p* < 0.001.

## Data Availability

Data supporting the reported results are contained within the article.

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
