# Peer review of "Developmental Neurotoxicity of Trichlorfon in Zebrafish Larvae"

_ijms, 2023, doi:10.3390/ijms241311099_

Round 1
Reviewer 1 Report (Previous Reviewer 2)
The edits were performed correctly.
Author Response
We are grateful to the reviewer's positive comments!
Reviewer 2 Report (New Reviewer)
The authors present a mechanistic study about the teratogenicity of the organophosphorus pesticide trichlorfon in zebrafish larvae. The authors conclude that this pesticide, allowed using in aquaculture, exerting developmental neurotoxicity through the induction of disorders in the cholinergic, dopaminergic and serotonergic signaling pathway. The results are not especially surprising since it is well known that organophosphorus pesticides inhibit acetylcholinesterase and therefore can cause imbalance in the neurotransmitter systems. Nevertheless, the study is well designed and the conclusions are solid and robust and therefore the manuscript would be publishable in IJMS after amendment of a few minor issues that I am relating in next paragraphs.
My main concern is about the maximum allowed trichlorfon concentration in aquaculture. It is stated different figures along different parts of the manuscript. It is not possible to be 0.1-1 mg/L up to 25 g/L as stated in lines 35-36. In line 235 the concentration seems to be 0.1 mg/L. Please, state clearly the maximum legally allowed trichlorfon concentration. It is hard to understand how trichlorfon concentrations causing acethylcholinesterase inhibitions higher than 50% and significant teratogenicity can be sustained in practical terms.
By the other hand, whether the maximum concentration is 0.1 mg/L it cannot be said, as in line 14, that this manuscript is using concentrations allowed in aquiculture because only the lowest concentration used here is allowed. Please, revise these parts of the text.
Lines 91-92 the percentages of survival rate (6.22) and hatching rate (7.26) are indeed reductions in these rates as regard control. The current text is confusing. Please, clarify.
In this same paragraph, the percentages of malformation rates (121.91 and 652.30) are unclear to which are referred.
I missed statistical analysis in Figure 2C. Are the differences in average speed statistically significant? According to the text in paragraph lines 103-113 it seems they are, but this is not stated in the plot.
Please, describe shortly the bases for the determination of neurotransmitters (section 4.6). Elisa determinations?
Round 2
Reviewer 2 Report (New Reviewer)
The authors have successfully addressed all my previous concerns. I have no further objections and suggest publication.
This manuscript is a resubmission of an earlier submission. The following is a list of the peer review reports and author responses from that submission.
Round 1
Reviewer 1 Report
Dear authors,
the work is very interesting but the data are reported in a non-exhaustive way. Statistical analyzes relating to malformations must be supported by at least representative images. Furthermore, to confirm the expressions relating to the cholinergic pathways, it would be advisable to insert positive controls, therefore molecules that have already been amply demonstrated to be pathway modulators, and a control using inhibitors in order to be able to appropriately evaluate the degree of neurotoxicity.
In materials and methods the reported protocol for RT-PCR is certainly wrong.
Furthermore, in line 31 it would be advisable to add the abbreviation TCF after the commercial name of the product.
Reviewer 2 Report
Over all the manuscript was written well and sufficient experiments were carried out. I just have few comments
Major comments
Line number 396: Larval samples were homogenized in 0.9% sodium chloride on ice. Since the half-life of these neurotransmitters are very less, especially 5-HT. I doubt if the lysate was prepared with saline, we could not estimate them. Kindly confirm are you using any other specific buffer or buffer provided in the Kit used for the estimation.
It is my suggestion to estimation the expression level of Tyrosine hydroxylase after trichlorfon treatment, as it is the key rate limiting enzyme for the synthesis of neurotransmitters like Dopamine and others.
Minor corrections
Line number 50: Brain is “one of the” potential target organs
Line number 61: Word assumed could be changed to “Hypothesized”
Line number 63: Therefore, we used zebrafish embryos, a “teratotoxicity” model to investigate….
Figure 1. B. Kindly use bright colours for the hydrogen band
Line number 108: reduced “for” the 5 mg/ml
Figure 2 Legend: Line number 116: exposed to 0, 0.1, 2 “and” 5 mg/ml
Line number 120 and 121: Compared to the control group at “each time point”
Line number 200: reference missing for that sentence.
Line number 245: Ach is involved in motor (“Voluntary movements”) and learning memory
Line number 316: “regulation of shha upon exposure to trichlorfon may affect the development of brain and other organs (Reference missing).
Line number 347: Embryos were exposed “at” 144hpf
Line number 349: larvae were removed “on” time.
Line number 354: Six larvae were selected randomly?
Round 2
Reviewer 2 Report
The questions and comments were carefully addressed by the authors.
Author Response
Thank for the reviewer's positive comments!